# Ciprofloxacin and Trimethoprim Adsorption/Desorption in Agricultural Soils

**DOI:** 10.3390/ijerph19148426

**Published:** 2022-07-10

**Authors:** Lucía Rodríguez-López, Vanesa Santás-Miguel, Raquel Cela-Dablanca, Avelino Núñez-Delgado, Esperanza Álvarez-Rodríguez, Paula Pérez-Rodríguez, Manuel Arias-Estévez

**Affiliations:** 1Soil Science and Agricultural Chemistry, Faculty of Sciences, University of Vigo, 32004 Ourense, Spain; vsantas@uvigo.es (V.S.-M.); paulaperezr@uvigo.es (P.P.-R.); mastevez@uvigo.es (M.A.-E.); 2Department of Soil Science and Agricultural Chemistry, Engineering Polytechnic School, University of Santiago de Compostela, 27002 Lugo, Spain; raquel.dablanca@usc.es (R.C.-D.); avelino.nunez@usc.es (A.N.-D.); esperanza.alvarez@usc.es (E.Á.-R.)

**Keywords:** emerging pollutants, antibiotics, crop soil, soil pollution

## Abstract

The current research focuses on the adsorption/desorption characteristics of the antibiotics ciprofloxacin (CIP) and trimethoprim (TRI) taking place in 17 agricultural soils, which are studied by means of batch-type experiments. The results show that adsorption was higher for CIP, with Freundlich K_F_ values ranging between 1150 and 5086 L^n^ µmol^1−n^ kg^−1^, while they were between 29 and 110 L^n^ µmol^1−n^ kg^−1^ in the case of TRI. Other parameters, such as the Langmuir maximum adsorption capacity (q_m(ads)_), as well as the K_d_ parameter in the linear model and also the adsorption percentages, follow the same trend as K_F_. Desorption was lower for CIP (with K_F(des)_ values in the range 1089–6234 L^n^ µmol^1−n^ kg^−1^) than for TRI (with K_F(des)_ ranging between 26 and 138 L^n^ µmol^1−n^ kg^−1^). The higher irreversibility of CIP adsorption was also confirmed by its lower n_F(des)_/n_F(ads)_ ratios, compared to TRI. Regarding soil characteristics, it was evidenced that nitrogen and carbon contents, as well as mineral fractions, had the highest influence on the adsorption/desorption process. These results can be considered relevant as regards the fate of both antibiotics when they reach the environment as pollutants and therefore could be considered in assessment procedures focused on environmental and public health aspects.

## 1. Introduction

Soils are complex and heterogeneous environment compartments, with multiple functions that depend on the processes taking place inside them, which influence the fate of compounds such as antibiotics reaching these media as contaminants. In fact, these antibiotics are considered within the group of emerging pollutants. In recent years, these substances have been detected in increasing concentrations in the environment, and it has been facilitated due to their increased use related to comorbidities associated with the COVID-19 pandemic [1]. According to Ezzariai et al. [2] it can be estimated that the amount of antibiotics used annually worldwide is between 100,000 and 200,000 t.

Between 30 and 90% of the amount of antibiotics ingested by humans is excreted in feces and urine, in their original form or as metabolites [1,3]. In wastewater treatment plants, antibiotics are not completely eliminated, with variate percentage removals, such as 13% in the case of TRI and 87% in the case of fluoroquinolones [4,5]. Although purification methods have been improved as regards antibiotics removal, their total elimination is not achieved, causing the study of adsorption methods using different materials such as polymers [6], activated carbon [7] or biochar [8] to still be clearly interesting and very abundant.

These emerging pollutants reach the soil after the use of wastewater for irrigation as well as due to the application of sludge as agricultural soil amendments [2], with both pathways being the main inputs of antibiotics into the environment, along with the application of slurries as soil amendments [9]. In this sense, in different European countries, including Spain, 50% of the sewage sludge produced is used as an agricultural amendment, since it has high organic matter and nutrient contents, including nitrogen and phosphorus [1,4]. Soils devoted to corn and vineyard production are among those most modified with sludge and manure to increase their fertility, since they are two of the ten most important products derived from agriculture in Spain [10].

Once antibiotics reach the soil, they can pose a serious risk to both human and ecological health. Among their associated environmental and public health risks, they can give rise to the appearance of resistant genes in bacteria, with relevant levels found in sediments, soils and waters [11]. Antibiotic resistance genes exist naturally in the chromosomes of bacteria present in different environmental compartments, but currently, due to the pressure exerted by the high presence of antibiotic pollutants, resistant genes are also found in plasmids [3], which can increase their transmission to other organisms, both pathogenic and non-pathogenic. In addition, different studies suggest that the combination of antibiotics with other contaminants, such as heavy metals, taking place in soils can favor the proliferation of antibiotic resistance [3,12]. An additional problem is the bioaccumulation of these compounds, which leads to high concentrations in various plants, favoring their entry into the food chain [13,14]. Antibiotics can also move through the soil and contaminate surface, subsurface and groundwater, causing increasingly high concentrations of these contaminants to be found in water bodies [15]. This depends on the characteristics of the specific antibiotic molecule, as well as on those of the soil and the environmental conditions [16]. Their fate will largely depend on the chemical form of the antibiotics present in the soil, since they are molecules which can behave as neutral and/or charged species (in the form of zwitterion, negatively or positively charged), and will also depend on the multiple processes that take place in the soil, such as degradation, transport (for example by runoff and leaching), plant uptake, as well as adsorption/desorption.

Among the most widely used antibiotics are ciprofloxacin (CIP), which belongs to the family of fluoroquinolones, and trimethoprim (TRI), a diaminopyrimidine, which are characterized by being broad-spectrum biocides. Fluoroquinolones are among the families most present in sewage sludge, along with tetracyclines and sulfonamides [2,17], with concentrations of ciprofloxacin found in soils reaching between 0.57 µg kg^−1^ and 0.35 mg kg^−1^ [9]. In the case of TRI, its presence in soils has been reported to be between 0.64 and 2.15 µg kg^−1^ [4]. The possibility for these antibiotics to reach water bodies, as well as crops, and the food chain, will be clearly affected by adsorption-desorption. A high adsorption and low desorption will favor the retention of these compounds in the soil, this process being conditioned by molecular characteristics of the antibiotics, as well as by soil physicochemical properties (such as pH, mineral concentration, cation exchange capacity, organic matter content and structure [16]).

In view of the above, the main objective of this study is to elucidate the main characteristics of the adsorption-desorption processes affecting CIP and TRI antibiotics which contact agricultural soils with different edaphic properties, since both antibiotics have not been widely studied in soils up to now. Therefore, this study was carried out to determine the probable time-course evolution of these antibiotics and their fate once they reach the environment as pollutants, taking into account that retention/release will be key as regards their mobility and its possible impact on water bodies, the food chain and ultimately on environmental and human health.

## 2. Materials and Methods

### 2.1. Soil Samples

In this study, 17 agricultural soils were used: 10 vineyard soils (soils 1–10) and 7 soils dedicated to corn cultivation (soils 11–17), located at different areas of the northwest of Spain, specifically in Galicia. These crops are two of the most widely cultivated in the world, and the soils that were sampled for this study were selected due to the variability in their pH values and organic matter contents. Soil samples were taken with an Edelman probe at a depth of 0–20 cm, air-dried, sieved by 2 mm and stored in polyethylene bottles until analysis. In each of the soils, the final sample resulted from mixing and homogenizing 10 subsamples, randomly taken in each respective area.

The particle size distribution was determined in the <2 mm fractions, by means of the pipette method, carried out after a previous treatment with H_2_O_2_ (6%) to eliminate organic matter, and with 0.1 M HCl to eliminate Fe and Al oxides [18]. As a result, 3 different fractions were quantified: sand (2–0.05 mm), silt (0.05–0.002 mm) and clay (<0.002 mm). Soil pH was measured in water and in 0.1 M KCl (using 1:2.5 as soil:solution ratio). Total organic carbon and total organic nitrogen were determined by means of an elemental analyzer (1112 Series NC, Thermo Finnigan, Amsterdam, The Netherlands).

The effective cation exchange capacity (eCEC) was estimated as the sum of the basic exchangeable cations (Na_e_, K_e_, Ca_e_ and Mg_e_), which were extracted with 0.2 M NH_4_Cl [19], and the exchangeable Al (Al_e_), which was extracted with 1 M KCl [20]. Ca, Mg and Al were quantified via flame atomic absorption spectrophotometry, while Na and K were determined via atomic emission spectrophotometry (with an AAnalyst 200 spectrophotometer, Perkin Elmer, Boston, MA, USA).

### 2.2. Chemicals and Reagents

The antibiotics ciprofloxacin (CIP) and trimethoprim (TRI) were supplied by Sigma-Aldrich (Barcelona, Spain), both with a purity of 98%. Their main characteristics are shown in Table 1. All the reagents used were of high purity analytical grade, supplied by Panreac (Barcelona, Spain) and by Fisher Scientific (Madrid, Spain) in the case of acetonitrile.

### 2.3. Adsorption/Desorption Experiments

To carry out adsorption experiments, aliquots of 0.5 g of soil were weighed in 50 mL centrifuge tubes (Deltalab, Spain) and suspended in 40 mL of solution of each of the antibiotics (CIP and TRI), at 7 different concentrations (between 2.5–50 µM for TRI and between 5–400 µM for CIP), all of these containing 0.005 M CaCl_2_ as the ionic background electrolyte.

The suspensions were shaken for 48 h in the dark, at 50 rpm, on a rotary shaker and at room temperature (25 ± 1 °C). Adsorption kinetic studies were previously carried out, indicating that 48 h is enough time to reach equilibrium. Once shaken, the samples were centrifuged for 15 min at 4000 rpm using a Rotina 35R centrifuge (Hettich Zentrifugen, Tuttlingen, Germany). Then, these samples were filtered using nylon syringe filters (0.45 µm pore size). Antibiotic concentrations were quantified via HPLC using 2 mL capacity Eppendorf propylene vials (Fisherbrand, Madrid, Spain). The pH of the samples was also measured using a combined glass micro-electrode (Crison, Barcelona, Spain). The amount of antibiotic adsorbed in the soils was calculated as the difference between the amount initially added and that remaining in the solution once equilibrium is reached (48 h).

To study desorption, the soil samples previously used in the adsorption experiments were weighed to determine the amount of solution left in the soil, then the soil was re-suspended in 40 mL of 0.005 M CaCl_2_. These samples were then shaken, centrifuged, filtered and analyzed in the same way as in the adsorption phase.

All determinations were made in triplicate.

### 2.4. Quantification of the Antibiotics CIP and TRI

The quantification of the antibiotics was performed following the methodology previously detailed in Rodríguez-López et al. [24]. Briefly, the analysis was carried out in Ultimate 3000 HPLC equipment (Thermo Fisher Scientific, Madrid, Spain), with a quaternary pump, an auto-sampler, a thermostatted column compartment and an Ultimate 3000 series UV detector. This equipment was connected to a computer with version 7 of the Chromeleon software (Thermo Fisher Scientific, Madrid, Spain). Chromatographic separations were carried out with a C18 analytical column (150 mm long; 4.6 mm internal diameter; 5 μm particle size) from Phenomenex (Madrid, Spain) and a security column (4 mm long; 3 mm internal diameter; 5 μm particle size), packed with the same material as the column.

The quantification limits were 0.1 µM for CIP and 0.05 µM for TRI while the detection limits were 0.03 µM and 0.015 µM, respectively. The injection volume was 50 µL and the flow rate was 1.5 mL min^−1^. Phase A was acetonitrile and phase B was 0.01 M phosphoric acid (pH = 2). The dilution gradient was from 5 to 32% for phase A and from 95 to 68% for phase B, in a time of 10.5 min. The initial conditions were resumed after 2 min, with a total analysis time of 15 min and retention times of 6.5 min for CIP and 5.6 min for TRI. The wavelength used was 212 nm for both antibiotics.

### 2.5. Statistical Analysis and Data Treatment

The data obtained in the adsorption and desorption experiments were described using the Freundlich, Langmuir and linear models, which correspond to the following equations, respectively:(1)$$q_{a}=K_{F}C_{eq}^{n}\;$$
(2)$$q_{a}=\frac{K_{L\;}C_{eq}q_{m}}{1+K_{L}C_{eq}}\;$$
(3)$$q_{a}=K_{d}C_{eq}\;$$
where q_a_ (µmol kg^−1^) is the amount of antibiotic adsorbed at equilibrium, C_eq_ (µmol L^−1^) is the concentration of antibiotic that remains in the equilibrium solution, K_F_ (L^n^ µmol^1−n^ kg^−1^) is the Freundlich affinity coefficient; n (dimensionless) is the Freundlich linearity index, K_L_ (L µmol^−1^) is the Langmuir constant related to the adsorption energy, q_m_ (µmol kg^−1^) is the maximum adsorption capacity of the soil and K_d_ (L kg^−1^) is the distribution coefficient in the linear model.

The adjustment of the different models to the experimental data was carried out using IBM SPSS Statistics 21.0 software (New York, NY, USA). In addition, bivariate Pearson correlations and multiple linear regressions were performed between the adsorption and desorption parameters and the different soil properties, using the same software.

## 3. Results

### 3.1. Soil Characteristics

Table 2 shows the physicochemical characteristics of the soils studied. These soils have a pH in water in the range 5.0 to 8.0 and a pH in KCl from 4.2 to 7.8. The lower values of pH in KCl with respect to those in water are indicative of a predominance of negative charge in these soils. Regarding eCEC values, they show high variability, ranging between 5.43 and 42.81 cmol(c)kg^−1^, with Ca being the predominant element in most cases. Soil organic carbon (SOC) also shows high variability, with values between 1.0 and 7.7%, while total soil nitrogen (TSN) has values between 0.11 and 0.63%. Sand is the predominant textural fraction in all soils, with percentages between 43 and 69%, followed by silt (between 10 and 34%) and clay (between 12 and 25%), with which the textures are sandy-loam (soils 3, 4, 9, 15 and 16), loam (soils 1, 2, 5, 11, 12 and 13) and sandy-clay-loam (soils 6, 7, 8, 10, 14 and 17).

### 3.2. CIP and TRI Adsorption

Figure 1 and Figure 2 show the adsorption curves for the antibiotics CIP and TRI, respectively, plotting the amount adsorbed in the soil (q_a_, in µmol kg^−1^) against the concentration present at equilibrium (C_eq_, in µmol L^−1^), for the 17 soils studied.

In the case of CIP (Figure 1), the curves are L-type (as per Giles at al. [25]), indicating that the antibiotic has a high affinity for the adsorption sites. These are non-linear and concave curves, which indicates that at low concentrations of antibiotic there is a high affinity for the soil, with almost all the amount of pollutant being adsorbed. In fact, the adsorption percentages are greater than 90% for concentrations added ranging from 5 to 100 µM, while adsorption percentages ranged between 56 and 97% for added concentrations going from 200 to 400 µM (Appendix A). Median values ranged from 97% to 79%, being slightly lower for added concentrations of 220–400 µM (Appendix A). These high adsorption percentages are a confirmation of the affinity of this antibiotic for the soils studied.

Table 3 shows that, judging by the R^2^ values, the fits were satisfactory for the three adsorption models assessed (Freundlich, with R^2^ values between 0.937 and 0.998, Langmuir, with R^2^ values between 0.946 and 0.998, and linear, with R^2^ values between 0.840 and 0.976). Among these R^2^ values, those corresponding to the Langmuir model are slightly higher, indicating that it is the model that best fits the experimental data. K_F_ values ranged between 1150 and 5086 L^n^ µmol^1−n^ kg^−1^, with a mean value of 3334 L^n^ µmol^1−n^ kg^−1^. The values of the n parameter were lower than 1, indicating a certain concavity of the adsorption curves. Maximum adsorption calculated using the Langmuir equation (q_m_) ranged between 17,264 and 40,722 µmol kg^−1^, with a mean value of 30,236 µmol kg^−1^, and K_L_ ranged between 0.01 and 0.14 L µmol^−1^. K_d_ values were lower than those of K_F_, ranging between 90 and 1322 L kg^−1^, with a mean score of 364 L kg^−1^ (Table 3).

Figure 2 shows that, in the case of TRI, adsorption curves have a higher tendency to linearity, although they can also be considered type L, but with a much lower slope compared to those of CIP. In fact, the fits are satisfactory for the three models, judging by the values of R^2^, and for CIP, the Langmuir model is the one that best describes the experimental data, since its R^2^ values are the highest.

Adsorption percentages in this case were much lower, with mean values of 22–40% and with a median of 20–38% for the different initial concentrations added (Appendix A). K_F_ values ranged between 29 and 125 L^n^ µmol^1−n^ kg^−1^, while the values of the n parameter were in this case closer to 1, ranging between 0.62 and 0.84. Regarding K_d_, its values were very similar to those of K_F_, ranging between 10 and 48 L kg^−1^. The values of n and K_d_ confirm the greater linearity in the adsorption curves corresponding to TRI, compared to those of CIP. The q_m_ values ranged between 1297 and 4460 µmol kg^−1^, with a mean of 2209 µmol kg^−1^, and those of K_L_ ranged between 0.007 and 0.036 L µmol^−1^, being lower than those obtained for CIP (Table 3).

### 3.3. Desorption of CIP and TRI

Appendix A show desorption curves for CIP and TRI, respectively, corresponding to the 17 soils under study. These figures plot the amount of CIP (Appendix A) and TRI (Appendix A) that remained adsorbed to the soil after a desorption cycle (q_m(des)_, µmol kg^−1^) against the concentration of CIP or TRI in the equilibrium solution (C_eq(des)_, µmol L^−1^).

For CIP, the slopes of the desorption curves were greater than those of adsorption, indicating that an important part of CIP remains adsorbed to the soil after a desorption cycle. In addition, it was noted that as the amount of CIP initially added was increased, desorption was higher (Table 4). The desorption percentages were low, in the range of 0.0–22.4%, with a mean value of 3.0–9.2% for all concentrations used (5–400 µM) and a median of 2.9–6.9% (Table 4).

Table 5 shows values of the desorption parameters related to the fits of the experimental data to the Freundlich, Langmuir and linear equations. The R^2^ values were above 0.921 for all the models, with the Freundlich model being the one that best fits the antibiotic CIP due to its higher values, while for TRI the best fit corresponded to the linear model. K_F(des)_ ranged between 1089 and 6234 L^n^ µmol^1−n^ kg^−1^, with a mean value of 3688 L^n^ µmol^1−n^ kg^−1^, which are similar to those obtained for adsorption (Table 3). The n_(des)_ values were higher than those of adsorption, in the range of 0.42–0.89. The q_m(des)_ parameter ranged between 18,154 and 99,941 µmol kg^−1^, with a mean of 40,902 µmol kg^−1^, while K_L_ ranged between 0.01 and 0.21 L µmol^−1^. Both parameters are indicative of low desorption taking place, since their values are similar to those of q_m_ and K_L_ obtained for adsorption. K_d_ values for desorption are higher than those of adsorption, ranging between 270 and 2337 L kg^−1^, with a mean of 946 L kg^−1^ (Table 5).

For TRI, K_F(des)_ ranged between 26 and 138 L^n^ µmol^1−n^ kg^−1^, while n_(des)_ values were in the range 0.60–1.28 (Table 5), scores similar to those obtained for adsorption. In addition, K_d_ values were also very similar to those of K_F_, ranging between 29 and 109 L kg^−1^. The values of q_m(des)_ ranged between 353 and 1558 µmol kg^−1^, with a mean of 903 µmol kg^−1^, which are lower than those obtained for adsorption. K_L_ ranged between 0.047 and 0.315 L µmol^−1^, which are higher than those obtained for adsorption. Both parameters indicate that the reversibility of adsorption is higher for TRI than for CIP.

Table 6 shows that, in the case of TRI, the slopes of the desorption curves were similar to those of adsorption, which indicates a higher reversibility in the adsorption process. This is confirmed by the fact that desorption percentages were higher for TRI than for CIP. In the case of TRI they were in the range 28.8–74.9%, with a mean value between 49.0 and 59.7% and a median similar to the mean (varying between 51.0 and 60.4%).

## 4. Discussion

This discussion will be carried out focusing on two fundamental aspects: on the one hand, the comparison of the adsorption/desorption results for CIP and TRI and, on the other hand, the relations between the adsorption/desorption parameters and the edaphic variables, studied by means of Pearson’s correlation and multiple regression analyses.

The values of the adsorption parameters indicate that CIP is more adsorbed than TRI, judging by its higher K_d_, K_F_ and q_m_ scores (Table 3). In addition, adsorption percentages are higher for CIP (Appendix A).

Different studies have shown a high variety of values for K_d_ for CIP, some of them being higher than those of the present work, as is the case of Leal et al. [26], who obtained K_d_ values in the range of 727–1277,874 L kg^−1^ for a set of 13 soils with different physical-chemical properties, or those provided by Conkle et al. [27], reaching 4844 L kg^−1^, as referenced in a review paper by Riaz et al. [9]. Values of the same order as those obtained in the present work (between 410–11,290 L kg^−1^) have also been reported by Uslu et al. [28], corresponding to three soils studied in Germany, or values of 430 L kg^−1^ also in soils from Germany [29,30], and between 300–45,000 L kg^−1^ reported by Vasudevan et al. [31], derived from a study of 30 soils in the USA.

K_F_ data are also reported in the literature. Specifically, Rath et al. [17] found values of 230–1366 mL^n^ µg^1−n^ g^−1^, which (although expressed in different units) are of a similar order to those obtained in the present work. These authors also observed that the adsorption of CIP is mainly due to the electrostatic interaction between the protonated part of the antibiotic and the negative charges of the soil. Other authors, such as Movasaghi et al. [32], who studied CIP adsorption in oat hulls, reported the influence of pH, and specifically that at low pH the positive charges of the antibiotic and the positive charges of the adsorbent surface give rise to a certain electrostatic repulsion, and therefore, adsorption is lower, while at a higher pH electrostatic attraction and greater adsorption take place. These authors also studied how mechanisms such as hydrogen bonding or electron donor acceptor (EDA) interactions can take place, since, for example, the organic compounds of the soil which have aromatic rings will present interactions as donor, with the benzene ring of CIP behaving as an acceptor. In the case of hydrogen bonds, it can also be one of the CIP adsorption mechanisms, since functional groups such as hydroxyl or carboxyl found on the surface of soil organic matter favor these bonds with carbonyl groups or/and hydroxyl groups of CIP [32,33]. In the case of soils with a pH around neutrality, the CIP molecule has a certain hydrophobicity, which leads to a low solubility of the antibiotic and therefore the adsorption process is facilitated, but these interactions may not explain the high adsorption that has been reported, since fluoroquinolones in general have low log K_OW_ values [28,32]. In the study by Movasaghi et al. [32], K_F_ values of 19,000–32,000 mL^n^ µg^1−n^ g^−1^ are referenced for oat hulls, while Sidhu et al. [34] reported K_d_ values of 357 L kg^−1^ for other adsorbents, such as bio-solids.

In addition, the fact of obtaining lower adsorption values for TRI than for CIP coincides with what is reported in the literature. Specifically, K_d(ads)_ values for TRI were in the range of 9–311 L kg^−1^ in different soils in Australia [35], while in Chinese soils they were in the range of 9.28–10.24 L kg^−1^ [36] and 5.88–21.8 L kg^−1^ [37]. Other researchers also found K_F(ads)_ values confirming the low levels of TRI adsorption [23,38,39]. However, some bio-adsorbents, such as activated carbon [7], biochard [8] and bentonite [40], show high TRI adsorption. The low adsorption of TRI in soils can be associated with the amphoteric nature of the molecule. Specifically, at pH 4–5 (which is the pH of the soils that are the object of this study), authors such as Peng et al. [37] observed that the TRI molecule behaves as anionic, which gives rise to an electrostatic repulsion with the negative charges of the soil, which makes adsorption relatively low, as confirmed by other studies, such as the one carried out by Kodesová et al. [23]. Other authors such as Berges et al. [7] concluded that TRI adsorption is due to π-π interactions between the two aromatic rings of the molecule and the surface of the adsorbate.

Comparing with other families of antibiotics, it has been reported that CIP (a fluoroquinolone) has high adsorption values, although lower than those of groups such as tetracyclines, showing K_F_ scores of up to 11,000 L^n^ µmol^1−n^ kg^−1^ (chlortetracycline), which have a high affinity for soils [41]. Furthermore, TRI, which has low adsorption compared to CIP, presents adsorption values similar to groups such as sulfonamides, with K_F_ lower than 22 L^n^ µmol^1−n^ kg^−1^ for sulfachloropyridazine [42], or to the group of beta-lactams, with antibiotics such as amoxicillin showing K_F_ below 150 L^n^ µmol^1−n^ kg^−1^ [43]. This indicates that the family of fluoroquinolones, together with that of tetracyclines, would be among those with the highest adsorption in soils, and the family of diaminopyrimidines, beta-lactams and sulfonamides, among those with low affinity for soils.

Regarding desorption, CIP desorption percentages were lower than those of TRI, and the desorption parameters are higher for CIP, also indicative of low desorption. These kind of results had been reported in previous studies, as in the case of Conkle et al. [27], with K_d(des)_ values between 2788 and 5431 cm^3^ g^−1^ for CIP, being slightly higher than those obtained in the present study, which may be due to the different edaphic characteristics of the soils used. In the case of K_F(des)_ values, Rath et al. [17] reported values in the range 537–3293 µg^1−1/n^ (cm^3^)^1/n^ g^−1^ for CIP, similar to the mean obtained in the current study.

As for TRI, its desorption percentages are higher, and its desorption parameter values are lower, compared to those of CIP. Regarding the Freundlich affinity coefficient, K_F(des)_, an average of 56.8 L kg^−1^ was obtained, while Peng et al. [37] reported values in the range 7.0–36.0 L kg^−1^ for CIP in three Chinese soils with different edaphic characteristics. Franklin et al. [39] reported values of 300–1700 µg^1−1/n^ L^n^ kg^−1^, which, despite being in different units, indicate a medium desorption. Regarding K_d(des)_, Zhang et al. [36] reported values between 12.5 and 15.0 L kg^−1^, confirming the low scores for that parameter, which correspond to a high desorption. It should be noted that, in general, desorption studies are fewer than those dealing with adsorption, for both antibiotics.

When studying the reversibility of adsorption through the n Freundlich parameter, using the expression n_(des)_/n_(ads)_, the values are much lower for CIP than for TRI, which indicates a greater irreversibility of adsorption in the case of CIP.

To evaluate to what extent the adsorption/desorption results of both antibiotics are related to the different physical-chemical characteristics of the soils, a statistical study was carried out, and as a result, Table 7 shows the correlation matrix relating edaphic variables with the adsorption parameters, while Table 8 shows the correlation matrix relating edaphic variables with desorption parameters.

Regarding the Freundlich parameters, K_F(ads)_ of CIP did not correlate significantly with any edaphic variable analyzed (Table 7), but n_(ads)_ was significantly and positively correlated with variables of the change complex, such as Ca_e_ (r = 0.640, *p* < 0.01), Mg_e_ (r = 0.540, *p* < 0.05) and eCEC (r = 0.484, *p* < 0.01), and also with variables related to soil organic matter, such as SOC (r = 0.738, *p* < 0.01) and TSN (r = 0.643, *p* < 0.01). Authors such as Rath et al. [17] and Vasudevan et al. [31] have obtained similar results regarding the influence of the cation exchange capacity on CIP adsorption, that is, a higher cation exchange capacity favors the adsorption of this compound in the soil. The K_L(ads)_ Langmuir parameter was positively correlated with the silt fraction (r = 0.632, *p* < 0.01) and negatively correlated with the sand fraction (r = −0.593, *p* < 0.05), while the adsorption maximum q_m(ads)_ was positively correlated with Mg_e_ (r = 0.519, *p* < 0.05), SOC (r = 0.729, *p* < 0.01) and TSN (r = 0.736, *p* < 0.01). Finally, K_d(ads)_ correlated positively with SOC (r = 0.605, *p* < 0.05) and with TSN (r = 0.708, *p* < 0.01). The correlations between the soil organic matter (SOC) and different adsorption parameters indicate that the adsorption onto this fraction is very important, as shown by authors such us Teixidó et al. [44] and Uslu and Yediler [28], who related this adsorption to the mechanisms of cation bridging, electrostatic interactions or hydrogen bonding.

From the results of the multiple regression analysis between CIP adsorption parameters and edaphic variables, the following considerations can be made:

(a) No equation with significant fit was found relating K_F(ads)_ and the soil variables analyzed.

(b) For Langmuir’s K_L(ads)_ and q_m(ads)_, the following significant equations were obtained:K_L(ads)_ = −0.005 ± 0.026 + Silt * 0.003 ± 0.001
R^2^ corrected: 0.360
F: 9.990, *p* = 0.006
q_m(ads)_ = 22,076 ± 2125 + TSN * 26937 ± 6393
R^2^ corrected: 0.512
F: 17.7540, *p* = 0.0008

(c) For K_d(ads)_, the following significant equation was obtained via linear regression fits:K_d(ads)_ = −57.3 ± 119.0 + TSN * 1390.5 ± 357.9
R^2^ corrected: 0.468
F: 15.095, *p* = 0.001

From these equations it can be stated that TSN is the edaphic variable most involved in CIP adsorption, explaining 51% of the variation of q_m(ads)_ and 47% of K_d(ads)_. On the other hand, the silt fraction explains 36% of the variation of K_L(ads)_.

Regarding TRI, K_F(ads)_ was significantly and positively correlated with SOC (r = 0.641, *p* < 0.01) and with TSN (r = 0.745, *p* < 0.01), which is different from CIP. In addition, n_(ads)_ was correlated with K_e_ (r = 0.557, *p* < 0.05), while Langmuir’s K_L(ads)_ was positively correlated with Ca_e_ (r = 0.558, *p* < 0.05) and negatively correlated with K_e_ (r = −0.552, *p* < 0.05), and the adsorption maximum q_m(ads)_ was positively correlated with Mg_e_ (r = 0.588, *p* < 0.05), K_e_ (r = 0.518, *p* < 0.05) and eCEC (r = 0.677, *p* < 0.01). K_d(ads)_ was only significantly and positively correlated with TSN (r = 0.579, *p* < 0.05). Peng et al. [37] reported that high organic matter contents and a high exchange capacity are positively related to a greater adsorption of TRI, as we found in the current work.

Using multiple regression analysis, the following significant equations were obtained for TRI:K_F(ads)_ = 16.3 ± 10.4 + TSN * 135.0 ± 31.3
R^2^ corrected: 0.525
F: 18.6, *p* < 0.0006
K_L(ads)_ = 0.0151 ± 0.0029 + Cae * 0.0007 ± 0.0003 + Ke * −0.0008 ± 0.0003
R^2^ corrected: 0.462
F: 7.4, *p* = 0.0070
q_m(ads)_ = 1243.0 ± 332.3 + eCEC * 70.0 ± 20.3
R^2^ corrected: 0.420
F: 11.9, *p* = 0.004
K_d(ads)_ = 8.98 ± 4.36 + TSN * 36.04 ± 13.10
R^2^ corrected: 0.291
F: 7.6, *p* = 0.015

Also for TRI, the edaphic variable TSN has relevance, explaining 52% of the variation of K_F(ads)_ and 29% of K_d(ads)_. Furthermore, in this case edaphic variables related to ion exchange are important, with Ca_e_ and K_e_ explaining 46% of the variation of K_L(ads)_, while eCEC explains 42% of q_m(ads)_.

CIP desorption parameters were only significantly correlated in the following cases (Table 8): K_L(des)_ correlated negatively with the sand fraction (r = −0.501, *p* < 0.05) and positively with the silt fraction (r = 0.553, *p* < 0.05); q_m(des)_ correlated positively with Al_e_ (r = 0.589, *p* < 0.05) and negatively with the silt fraction (r = −0.494, *p* < 0.05); finally, K_d(des)_ was positively correlated with SOC (r = 0.499, *p* < 0.05) and TSN (r = 0.621, *p* < 0.01).

For CIP desorption, the following significant equations were obtained:K_L(des)_ = −0.0051 ± 0.0451 + Silt * 0.0047 ± 0.0018
R^2^ corrected: 0.260
F: 6.6, *p* = 0.021
q_m(des)_ = 30,710 ± 6088 + Al_e_ * 46893 ± 16592
R^2^ corrected: 0.304
F: 8.0, *p* = 0.013
K_d(des)_ = 305 ± 229 + TSN * 2114 ± 689
R^2^ corrected: 0.345
F: 9.4, *p* = 0.008

As in the case of adsorption, CIP desorption is fundamentally affected by TSN, which explains 34% of K_d(des)_, while the silt fraction explains 26% of the variation in K_L(des)_, and Al_e_ explains 30% of the variation of q_m(des)_.

In the case of TRI desorption, a significant and positive correlation of K_F(des)_ with the sand fraction was found (r = 0.508, *p* < 0.05) and a negative correlation with the silt fraction (r = −0.502, *p* < 0.05). The parameter n_(des)_ was positively correlated with pH in water (r = 0.482, *p* < 0.05) and with pH in KCl (r = 0.611, *p* < 0.01), as well as with exchange parameters such as Ca_e_ (r = 0.712, *p* < 0.01), Mg_e_ (r = 0.640, *p* < 0.01) and eCEC (r = 0.595, *p* < 0.05). Langmuir’s K_L(des)_ was negatively correlated with Mg_e_ (r = −0.664, *p* < 0.01), while q_m(des)_ was positively correlated with Ca_e_ (r = 0.625, *p* < 0.05), Mg_e_ (r = 0.852, *p* < 0.01) and eCEC (r = 0.845, *p* < 0.01), as well as with SOC (r = 0.614, *p* < 0.05) and TSN (r = 0.660, *p* < 0.05). K_d(des)_ was positively correlated with SOC (r = 0.751, *p* < 0.05) and TSN (r = 0.725, *p* < 0.05), and also with the sand fraction (r = 0.517, *p* < 0.05), and negatively correlated with the clay fraction (r = −0.554, *p* < 0.05).

Through multiple regression analysis, using the TRI desorption parameters as dependent variables, the following significant equations were obtained:K_F(des)_ = −7.50 ± 38.15 + Sand * 1.52 ± 0.67
R^2^ corrected: 0.208
F: 5.2, *p* = 0.037
K_L(des)_ = 0.221 ± 0.038 + Mg_e_ * − 0.121 ± 0.041
R^2^ corrected: 0.390
F: 8.7, *p* = 0.013
q_m(des)_ = 265.1 ± 132.1 + Mg_e_ * 770.7 ± 142.6
R^2^ corrected: 0.702
F: 29.2, *p* = 0.0002
K_d(des)_ = 22.13 ± 7.93 + SOC * 8.48 ± 1.93
R^2^ corrected: 0.534
F: 19.3501, *p* = 0.0005

As was the case for adsorption, the most relevant variables related to TRI desorption are some grain size fractions (specifically the sand fraction, which explains 21% of K_F(des)_ variation), and variables related to the exchange complex (such as Mg_e_, which explains 39% of the variation of K_L(des)_ and 70% of the variation of q_m(des)_), and, finally, SOC, which explains 53% of the variation of K_d(des)_.

## 5. Conclusions

The antibiotics included in this study show a clearly differentiated adsorption/desorption behavior in the soils under examination. Specifically, on the one hand, CIP shows more affinity (56–100%) for the 17 agricultural soils used than TRI, which shows lower adsorption (13.3–74.3%). In the case of the data obtained for desorption, the behavior of both antibiotics is also very different, the values of TRI are clearly higher (28.8–74.9%) than the values of desorption of CIP (0.0–22.4%). The results of both antibiotics fit satisfactorily to the three models used, Freundlich, Langmuir and linear, with values of the different constants being similar to those obtained in previous studies. Overall, the adsorption/desorption process presents high irreversibility for CIP, contrary to what happens for TRI. The edaphic properties that most condition the adsorption processes are carbon and nitrogen contents, in the case of CIP, while they are nitrogen, potassium and cation exchange capacity when focusing on TRI. Regarding desorption, in the case of TRI it depends largely on nitrogen content, cation exchange capacity and soil texture, while for CIP it is largely dependent on silt, magnesium and nitrogen contents. These results are relevant, since determining the variables that influence the adsorption/desorption of both antibiotics, as well as the reversibility of the adsorption process, facilitates taking appropriate decisions to face contamination due to these antibiotics affecting agricultural soils. In fact, it would be key to know what parameters can favor the adsorption of these pollutants and which alternatives could be appropriate to achieve their retention/removal, preventing transfer to other environmental compartments, such as water bodies, as well as their entry into the food chain. With this in mind, the set of results provided by this research can be considered relevant in terms of risk assessment, regarding environmental and public health aspects linked to pollution of agricultural areas that receive the application of solid materials (such as organic fertilizers) or liquids (such as sewage) containing these antibiotics. For future studies, it would be interesting to include additional soils, with characteristics clearly different from those evaluated in this research, as well as to evaluate the possible mitigating or remedial effect of adding low-cost bio-adsorbents (and especially by-products) to agricultural soils susceptible to this type of contamination.

## Figures and Tables

**Figure 1 ijerph-19-08426-f001:**
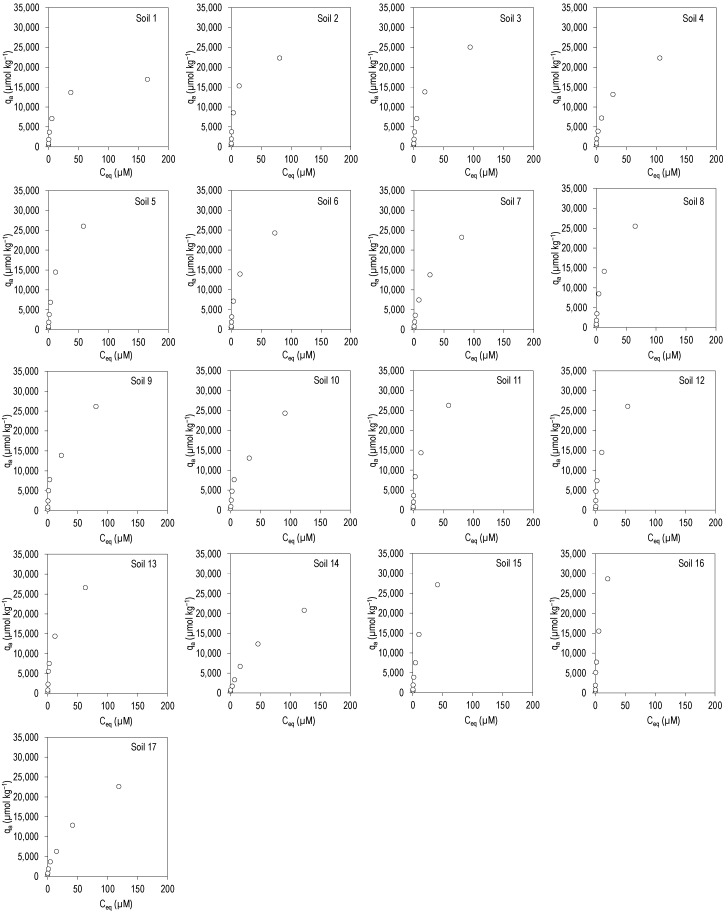
Adsorption curves for CIP in the 17 soils studied. q_a_: CIP adsorbed onto the soil; C_eq_: CIP concentration in the equilibrium solution. Average values (*n* = 3), with coefficients of variation always lower than 5%.

**Figure 2 ijerph-19-08426-f002:**
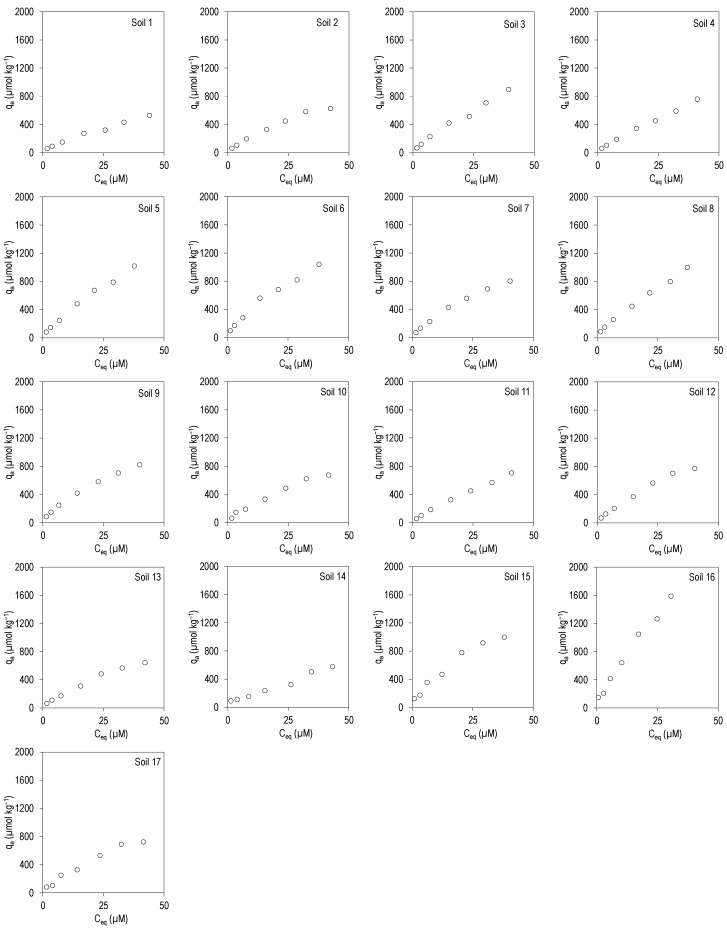
Adsorption curves for TRI in the 17 soils studied. q_a_: TRI adsorbed onto the soil; C_eq_: TRI concentration in the equilibrium solution. Average values (*n* = 3), with coefficients of variation always lower than 5%.

**Table 1 ijerph-19-08426-t001:** Main characteristics of the antibiotics studied. K_OW:_ coefficient of partition octanol/water.

Common Name	Chemical Structure	Chemical Formula	Molecular Weight (g mol^−1^)	Log K_OW_	pK_a_	Water Solubility (mg L^−1^)
Ciprofloxacin ^1^	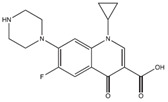	C_17_H_18_FN_3_O_3_	331.34	0.28	6.09–8.74	36,000
Trimethoprim ^2^	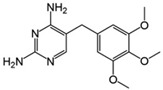	C_14_H_18_N_4_O_3_	290.32	0.91	6.16–7.16	400

^1^ [17,21]; ^2^ [22,23].

**Table 2 ijerph-19-08426-t002:** Main physicochemical properties of the soils studied. pH_w_: pH in water; pH_KCl_: pH in 0.1 M KCl; Ca_e_: exchangeable calcium; Mg_e_: exchangeable magnesium; Na_e_: exchangeable sodium; K_e_: exchangeable potassium; Al_e_: exchangeable aluminum; eCEC: effective cation exchange capacity; TSN: total soil nitrogen; SOC: soil organic carbon.

Soil	pH_w_	pH_KCl_	Ca_e_	Mg_e_	Na_e_	K_e_	Al_e_	eCEC	SOC	TSN	Sand	Silt	Clay
			cmol_(c)_ kg^−1^						%				
**1**	5.7	5.0	3.15	0.32	1.48	0.43	<0.1	5.43	1.0	0.11	48	30	22
**2**	5.6	4.8	3.38	0.26	1.83	0.50	<0.1	6.02	1.6	0.18	44	34	23
**3**	6.1	5.3	6.33	1.48	18.00	11.40	<0.1	37.23	2.8	0.23	61	21	18
**4**	5.6	4.9	4.74	1.07	1.64	11.91	0.16	19.52	3.7	0.26	69	17	14
**5**	6.1	5.5	8.43	1.56	2.46	10.12	<0.1	22.58	4.4	0.37	50	30	20
**6**	5.7	4.9	13.40	1.09	2.32	0.55	<0.10	17.36	4.8	0.40	58	20	21
**7**	5.5	4.2	6.55	0.37	2.60	0.27	0.72	10.51	6.6	0.49	66	13	21
**8**	7.1	6.5	8.11	0.51	0.08	0.37	<0.1	9.09	2.4	0.14	67	10	22
**9**	7.3	6.5	10.53	1.04	0.44	0.27	<0.1	12.32	3.6	0.22	67	20	12
**10**	5.8	4.8	5.73	0.83	0.17	1.13	0.74	8.60	4.0	0.30	62	18	20
**11**	5.0	4.8	4.58	0.67	0.10	0.17	0.43	5.94	2.0	0.21	46	32	22
**12**	5.7	5.2	6.18	0.77	−0.01	0.28	0.24	7.47	2.6	0.25	44	34	22
**13**	5.3	5.1	5.35	0.67	0.02	0.44	0.40	6.88	2.7	0.27	43	32	25
**14**	8.0	7.8	39.44	2.07	0.60	0.69	<0.1	42.81	5.0	0.33	50	28	23
**15**	6.4	6.5	21.65	1.60	0.16	0.93	<0.1	24.35	5.6	0.53	59	24	17
**16**	6.1	6.0	18.81	1.60	0.08	0.33	<0.1	20.84	7.7	0.63	63	24	12
**17**	5.4	5.1	4.85	0.39	0.13	0.38	0.89	6.63	3.2	0.23	61	18	21

**Table 3 ijerph-19-08426-t003:** Minimum, maximum, mean and median of the parameters resulting from the fits of the experimental adsorption data to the Freundlich, Langmuir and linear models, for ciprofloxacin (CIP) and trimethoprim (TRI). K_F_ (L^n^ µmol^1−n^ kg^−1^) is the Freundlich affinity coefficient; n (dimensionless) is the Freundlich linearity index; K_L_ (L µmol^−1^) is the Langmuir constant related to adsorption energy; q_m_ (µmol kg^−1^) is the maximum soil adsorption capacity; K_d_ (L kg^−1^) is the distribution constant in the linear model; R^2^: coefficient of determination.

**CIP**								
	**Freundlich**			**Langmuir ***			**Linear**	
	**K_F(ads)_**	**n_(ads)_**	**R^2^**	**K_L(ads)_**	**q_m(ads)_**	**R^2^**	**K_d(ads)_**	**R^2^**
Minimum	1150	0.33	0.937	0.010	17,264	0.946	90	0.840
Maximum	5086	0.60	0.998	0.140	40,722	0.998	1322	0.976
Mean	3334	0.48	0.983	0.073	30,236	0.986	364	0.937
Median	3422	0.47	0.988	0.080	29,985	0.992	297	0.938
**TRI**								
	**Freundlich**			**Langmuir**			**Linear**	
	**K_F(ads)_**	**n_(ads)_**	**R^2^**	**K_L(ads)_**	**q_m(ads)_**	**R^2^**	**K_d(ads)_**	**R^2^**
Minimum	29	0.62	0.963	0.007	1297	0.983	10	0.978
Maximum	125	0.84	0.999	0.036	4460	0.999	48	0.999
Mean	57	0.74	0.991	0.019	2209	0.991	19	0.991
Median	52	0.74	0.993	0.019	1818	0.994	19	0.991

* Soil 14 did not fit significantly and it was not considered.

**Table 4 ijerph-19-08426-t004:** Ciprofloxacin (CIP) desorbed in the equilibrium, expressed in µmol kg^−1^ (and in percentage into brackets) for each of the initial concentrations added (C_0_) and for each of the 17 soils studied. Mean, median, maximum and minimum refer to desorption percentages. nd: not determined.

	C_0_ (µmol L^−1^)						
Soil	5	10	25	50	100	200	400
1	12.0 (2.5)	17.5 (2.1)	37.5 (2.1)	96.0 (2.6)	327.1 (4.6)	1093.6 (8.1)	2077.1 (12.3)
2	10.6 (2.2)	12.6 (1.6)	24.8 (1.3)	49.5 (1.3)	235.1 (2.7)	556.1 (3.7)	1181.3 (5.3)
3	13.5 (3.0)	14.6 (1.9)	26.8 (1.5)	82.8 (2.2)	270.1 (3.8)	754.7 (5.5)	2228.4 (8.9)
4	12.5 (2.9)	21.0 (2.5)	47.8 (2.4)	201.1 (5.2)	380.1 (5.3)	855.7 (6.5)	1527.6 (6.9)
5	12.9 (2.7)	14.0 (1.8)	24.0 (1.3)	60.2 (1.6)	160.4 (2.3)	533.8 (3.7)	1766.1 (6.8)
6	11.8 (2.5)	12.4 (1.6)	22.4 (1.2)	40.8 (1.3)	178.7 (2.5)	472.6 (3.4)	1520.8 (6.2)
7	13.4 (2.8)	17.6 (2.2)	54.5 (2.8)	99.9 (2.8)	337.7 (4.5)	784.0 (5.7)	1240.1 (5.4)
8	12.9 (3.0)	16.2 (1.9)	28.3 (1.6)	79.9 (2.3)	275.6 (3.3)	633.0 (4.4)	2267.5 (9.0)
9	12.7 (3.6)	19.4 (2.2)	30.1 (1.2)	69.1 (1.4)	197.1 (2.5)	997.7 (7.2)	4056.7 (15.5)
10	11.7 (3.4)	25.9 (3.0)	95.2 (3.8)	156.2 (3.3)	359.0 (4.7)	1339.0 (10.3)	3969.2 (16.3)
11	11.4 (2.6)	13.7 (1.8)	25.9 (1.3)	43.9 (1.2)	163.9 (2.0)	573.9 (4.0)	1632.9 (6.2)
12	11.4 (3.2)	14.6 (1.7)	30.5 (1.3)	69.7 (1.5)	121.8 (1.6)	526.2 (3.6)	1779.1 (6.7)
13	15.6 (4.2)	19.5 (2.3)	29.0 (1.3)	57.7 (1.0)	120.8 (1.6)	570.6 (4.0)	2691.4 (10.1)
14	26.2 (6.1)	44.9 (5.9)	189.9 (11.3)	402.2 (12.2)	911.0 (13.7)	2215.2 (18.0)	4668.0 (22.4)
15	12.8 (3.0)	15.8 (2.0)	32.4 (1.7)	85.0 (2.2)	268.9 (3.6)	536.8 (3.7)	1744.0 (6.4)
16	11.4 (2.5)	13.1 (1.6)	18.2 (0.9)	53.9 (1.0)	100.7 (1.3)	309.1 (2.0)	904.1 (3.2)
17	0.0 (0.0)	42.1 (5.6)	134.5 (7.4)	nd	462.1 (7.4)	1033.0 (8.0)	2078.9 (9.2)
Mean	3.0	2.5	2.6	3.0	4.0	6.0	9.2
Median	2.9	2.0	1.5	2.2	3.3	4.4	6.9
Minimum	0.0	1.6	0.9	1.0	1.3	2.0	3.2
Maximum	6.1	5.9	11.3	12.2	13.7	18.0	22.4

**Table 5 ijerph-19-08426-t005:** Minimum, maximum, mean and median of the parameters resulting from the fits of the experimental desorption data to the Freundlich, Langmuir and linear models, for ciprofloxacin (CIP) and trimethoprim (TRI). K_F(des)_ (L^n^ µmol^1−n^ kg^−1^) is the Freundlich affinity coefficient; n_(des)_ (dimensionless) is the Freundlich linearity index; K_L(des)_ (L µmol^−1^) is the Langmuir constant related to adsorption energy; q_m(des)_ (µmol kg^−1^) is the maximum soil adsorption capacity; K_d(des)_ (L kg^−1^) is the distribution constant in the linear model; R^2^: coefficient of determination.

**CIP**								
	**Freundlich**			**Langmuir ***			**Linear**	
	**K_F(des)_**	**n_(des)_**	**R^2^**	**K_L(des)_**	**q_m(des)_**	**R^2^**	**K_d(des)_**	**R^2^**
Minimum	1089	0.42	0.966	0.010	18,154	0.958	270	0.921
Maximum	6234	0.89	0.999	0.210	99,941	0.999	2336	0.999
Mean	3688	0.60	0.989	0.106	40,902	0.989	946	0.962
Median	4029	0.56	0.990	0.120	31,691	0.992	964	0.960
**TRI**								
	**Freundlich**			**Langmuir**			**Linear**	
	**K_F(des)_**	**n_(des)_**	**R^2^**	**K_L(des)_**	**q_m(des)_**	**R^2^**	**K_d(des)_**	**R^2^**
Minimum	26	0.60	0.951	0.047	353	0.954	29	0.968
Maximum	138	1.28	0.998	0.315	1558	0.999	109	0.999
Mean	78	0.83	0.983	0.120	903	0.985	54	0.987
Median	78	0.82	0.988	0.091	836	0.988	48	0.988

* Soils 3, 14, 16 and 17 did not fit significantly and it was not considered.

**Table 6 ijerph-19-08426-t006:** Trimethoprim (TRI) desorbed in the equilibrium, expressed in µmol kg^−1^ (and in percentage into brackets) for each of the initial concentrations added (C_0_) and for each of the 17 soils studied. Mean, median, maximum and minimum refer to desorption percentages.

	C_0_ (µmol L^−1^)						
Soil	2.5	5	10	20	30	40	50
1	30.3 (54.4)	49.6 (54.0)	89.3 (59.9)	171.0 (69.8)	223.5 (70.3)	289.1 (67.1)	373.3 (70.6)
2	20.7 (35.1)	43.2 (41.3)	89.5 (46.0)	170.6 (52.2)	243.8 (54.6)	345.8 (59.5)	380.4 (61.0)
3	35.4 (55.9)	61.3 (51.1)	118.3 (52.1)	221.2 (52.7)	323.4 (63.0)	421.3 (59.7)	526.2 (58.9)
4	21.0 (35.0)	42.7 (40.6)	81.3 (43.3)	167.9 (49.2)	234.6 (52.0)	305.5 (52.0)	400.4 (52.8)
5	33.8 (41.4)	65.5 (45.1)	125.8 (50.7)	239.3 (49.6)	342.1 (50.8)	439.9 (55.7)	549.2 (53.9)
6	35.2 (36.5)	69.1 (40.5)	129.8 (46.4)	255.8 (50.4)	344.2 (50.7)	461.1 (56.6)	573.2 (55.3)
7	21.1 (28.8)	43.6 (32.8)	92.8 (40.6)	172.9 (40.4)	253.2 (45.5)	332.2 (48.1)	429.2 (51.0)
8	35.0 (42.0)	65.3 (44.5)	125.1 (48.8)	237.3 (53.3)	353.9 (55.7)	466.1 (58.5)	564.3 (60.4)
9	44.4 (53.2)	80.7 (54.7)	141.4 (56.8)	238.8 (57.5)	350.6 (60.6)	431.9 (61.4)	526.7 (64.3)
10	39.9 (64.8)	83.5 (59.6)	114.9 (60.2)	218.4 (65.9)	306.3 (62.9)	413.3 (66.4)	450.7 (66.8)
11	31.6 (53.4)	55.8 (53.9)	107.7 (57.7)	204.0 (63.0)	304.5 (67.2)	392.3 (69.0)	502.5 (71.2)
12	45.0 (65.6)	76.5 (59.5)	126.6 (61.7)	242.2 (65.2)	333.5 (59.1)	453.5 (64.6)	508.6 (65.8)
13	40.6 (63.8)	69.8 (62.8)	114.2 (66.2)	203.9 (65.4)	303.2 (62.8)	375.4 (66.4)	433.6 (67.6)
14	65.2 (71.4)	73.9 (66.7)	111.2 (71.4)	172.2 (72.7)	240.5 (74.9)	326.8 (65.0)	382.1 (66.5)
15	44.1 (35.3)	104.0 (59.8)	166.7 (47.0)	247.8 (52.9)	391.5 (50.3)	494.5 (53.9)	556.6 (55.8)
16	77.0 (51.0)	123.5 (59.2)	177.9 (42.8)	280.2 (43.6)	433.1 (41.2)	551.6 (43.8)	689.2 (43.4)
17	35.9 (44.7)	47.3 (45.1)	128.1 (51.6)	166.1 (58.8)	270.8 (51.3)	341.3 (49.6)	363.2 (50.4)
Mean	49.0	51.2	53.1	56.6	57.2	58.7	59.7
Median	51.0	53.9	51.6	53.3	55.7	59.5	60.4
Minimum	28.8	32.8	40.6	40.4	41.2	43.8	43.4
Maximum	71.4	66.7	71.4	72.7	74.9	69.0	71.2

**Table 7 ijerph-19-08426-t007:** Relations among adsorption parameters for ciprofloxacin (CIP) and trimethoprim (TRI) and edaphic variables (n = 17). K_F(ads)_ (L^n^ µmol^1−n^ kg^−1^) is the Freundlich affinity coefficient; n_(ads)_ (dimensionless) is the Freundlich linearity index; K_L(ads)_ (L µmol^−1^): Langmuir parameter related to adsorption energy; q_m(ads)_ (µmol kg^−1^): maximum adsorption capacity; K_d(ads)_ (L kg^−1^): distribution constant; pH_w_: pH in water; pH_KCl_: pH in 0.1 M KCl; Ca_e_: exchangeable calcium; Mg_e_: exchangeable magnesium; Na_e_: exchangeable sodium; K_e_: exchangeable potassium; Al_e_: exchangeable aluminum; eCEC: effective cation exchange capacity; TSN: total soil nitrogen; SOC: soil organic carbon.

CIP					
Variable	K_F(ads)_	n_(ads)_	K_L(ads)_	q_m(ads)_	K_d(ads)_
pH_H2O_	−0.280	0.351	−0.281	0.193	0.009
pH_KCl_	−0.174	0.458	−0.203	0.375	0.194
Ca_e_	−0.296	**0.640 ****	−0.307	0.464	0.255
Mg_e_	−0.107	**0.540 ***	−0.289	**0.519 ***	0.371
Na_e_	−0.071	−0.142	−0.145	−0.125	−0.179
K_e_	−0.128	−0.051	−0.230	−0.075	−0.154
Al_e_	−0.405	0.290	−0.462	0.158	−0.205
eCEC	−0.333	**0.484 ***	−0.418	0.332	0.095
SOC	−0.115	**0.738 ****	−0.379	**0.729 ****	**0.605 ***
TSN	0.080	**0.643 ****	−0.178	**0.736 ****	**0.708 ****
Sand	−0.349	0.411	**−0.593 ***	0.326	0.107
Silt	0.446	−0.365	**0.632 ****	−0.218	0.081
Clay	−0.090	−0.270	0.177	−0.393	−0.479
**TRI**					
pH_H2O_	0.110	0.026	0.198	0.357	0.186
pH_KCl_	0.211	0.040	0.219	0.482	0.287
Ca_e_	0.292	−0.053	**0.558 ***	0.442	0.279
Mg_e_	0.321	0.175	0.075	**0.588 ***	0.438
Na_e_	−0.164	0.322	−0.335	0.307	0.032
K_e_	−0.234	**0.557 ***	**−0.552 ***	**0.518 ***	0.085
Al_e_	−0.179	−0.193	−0.376	−0.386	−377
eCEC	0.099	0.291	−0.088	**0.677 ****	0.283
SOC	**0.641 ****	−0.191	0.373	0.168	0.458
TSN	**0.745 ****	−0.276	0.439	0.199	**0.579 ***
Sand	0.320	−0.021	0.034	0.312	0.255
Silt	−0.192	0.035	−0.014	−0.251	−0.155
Clay	−0.465	−0.002	−0.064	−0.266	−0.358

Significant at: * *p* < 0.05; ** *p* < 0.01.

**Table 8 ijerph-19-08426-t008:** Relations among desorption parameters for ciprofloxacin (CIP) and trimethoprim (TRI) and edaphic variables (n = 17). K_F(des)_ (L^n^ µmol^1−n^ kg^−1^) is the Freundlich affinity coefficient; n_(des)_ (dimensionless) is the Freundlich linearity index; K_L(des)_ (L µmol^−1^): Langmuir parameter related to adsorption energy; q_m(des)_ (µmol kg^−1^): maximum adsorption capacity; K_d(des)_ (L kg^−1^): distribution constant; pH_w_: pH in water; pH_KCl_: pH in 0.1 M KCl; Ca_e_: exchangeable calcium; Mg_e_: exchangeable magnesium; Na_e_: exchangeable sodium; K_e_: exchangeable potassium; Al_e_: exchangeable aluminum; eCEC: effective cation exchange capacity; TSN: total soil nitrogen; SOC: soil organic carbon.

CIP					
Variable	K_F(des)_	n_(des)_	K_L(des)_	q_m(des)_	K_d(des)_
pH_H2O_	−0.189	−0.156	−0.153	−0.319	−0.350
pH_KCl_	−0.087	−0.133	−0.084	−0.321	−0.235
Ca_e_	−0.193	0.105	−0.257	−0.154	−0.038
Mg_e_	−0.009	0.017	−0.194	−0.212	0.079
Na_e_	−0.088	0.042	−0.179	−0.015	−0.009
K_e_	−0.189	0.226	−0.307	0.158	0.022
Al_e_	−0.397	0.420	−0.382	**0.589 ***	−0.126
eCEC	−0.273	0.198	−0.413	−0.068	−0.026
SOC	−0.009	0.373	−0.303	0.321	**0.499 ***
TSN	0.172	0.285	−0.146	0.240	**0.621 ****
Sand	−0.317	0.400	**−0.501 ***	0.466	0.104
Silt	0.418	−0.414	**0.553 ***	**−0.494 ***	0.034
Clay	−0.122	−0.116	0.075	−0.145	−0.354
**TRI**					
pH_H2O_	−0.244	**0.482 ***	−0.268	0.239	−0.042
pH_KCl_	−0.354	**0.611 ****	−0.393	0.363	0.021
Ca_e_	−0.237	**0.712 ****	−0.389	**0.625 ***	0.173
Mg_e_	−0.182	**0.640 ****	**−0.664 ***	**0.852 ****	0.274
Na_e_	0.054	−0.053	0.350	0.049	−0.009
K_e_	0.137	0.004	−0.294	0.531	0.145
Al_e_	0.018	0.014	0.001	−0.160	0.069
eCEC	−0.131	**0.595 ***	−0.457	**0.845 ****	0.207
SOC	0.440	0.388	−0.278	**0.614 ***	**0.751 ****
TSN	0.437	0.301	−0.264	**0.660 ***	**0.725 ****
Sand	**0.508 ***	−0.009	−0.181	0.220	**0.517 ***
Silt	**−0.502 ***	0.075	0.114	−0.123	−0.377
Clay	−0.291	−0.097	0.302	−0.345	**−0.554 ***

Significant at: * *p* < 0.05; ** *p* < 0.01.

## Data Availability

Not applicable.

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
