# Peer review of "Ciprofloxacin and Trimethoprim Adsorption/Desorption in Agricultural Soils"

_ijerph, 2022, doi:10.3390/ijerph19148426_

Round 1
Reviewer 1 Report
This paper investigated the adsorption/desorption characteristics of the antibiotics Ciprofloxacin and Trimethoprim in 17 agricultural soils. Although the study conducted many tests related to antibiotics adsorption/desorption, the mechanism explanation was not clear and the results should also be discussed in-depth.
Please carefully read the following suggestions to improve the quality of the manuscript:
1. Title: It is hard to see your innovations. The core innovations were the adsorption/desorption behaviors of the antibiotics CIP and TRI. What did “retention” mean in your paper? What was the relationship between “retention” and “adsorption/desorption”. Please rearrange.
2. Key words: Do not just repeat the title. Please rearrange.
3. Introduction: (1) There were too many paragraphs in this section, and the logical structure needed to be reorganized. (2) Please add to previous research on the environmental processes of CIP and TRI in soils. (3) The innovations of this paper should be clarified in the introduction section. (4) Line 42-47, it was not clear that how antibiotics reached into the soil. The sludge containing antibiotics was used directly for soil amendment? Livestock raising might be an even bigger source of antibiotic pollution?
4. Materials and methods: (1) It was needed to explain how the sampling points of the 17 agricultural soils were selected. What were the representative of these soils? (2) It should be clarified that were there CIP and TRI detected in the soil sample. (3) Line 139-153, the author should also provide the relevant content of Quality Control, e.g., recoveries, method detection limits……
5. Results: (1) Figures and Tables should be redrew, and it was hard to observe the differences between different soils or different antibiotics in current version. (2) The paper applied Freundlich, Langmuir and Linear models to describe the adsorption data of CIP and TRI. Which model had the better performance? Why?
6. Discussion: (1) Current version was hard to understand the mechanisms of adsorption/desorption. It seemed that only the electrostatic interaction was discussed in the Discussion section. How about other adsorption mechanisms? e.g., π-π electron donor-acceptor (EDA) interactions, H-bonding, and electrostatic effects. This section should be improved. (2) The results of multiple regression analysis did not perform well, and the R2 values were very low. Other statistical models should be applied to further discuss these results. (3) Some soil physicochemical properties (pH, TSN, Al……) were shown to be correlated to antibiotic adsorption. It should be further discussed the mechanisms?
7. Writing norms of academic papers should be rule and standard. For example, the parameters format was not unified, and the use of italic was improperly in this paper.
Author Response
This paper investigated the adsorption/desorption characteristics of the antibiotics Ciprofloxacin and Trimethoprim in 17 agricultural soils. Although the study conducted many tests related to antibiotics adsorption/desorption, the mechanism explanation was not clear and the results should also be discussed in-depth.
Please carefully read the following suggestions to improve the quality of the manuscript:
- Title: It is hard to see your innovations. The core innovations were the adsorption/desorption behaviors of the antibiotics CIP and TRI. What did “retention” mean in your paper? What was the relationship between “retention” and “adsorption/desorption”? Please rearrange.
Response:
The title has been modified taking into account the comments from the Reviewer. On the other hand, a paragraph has been included clarifying the relationship between adsorption/desorption and retention (lines 81-85).
- Key words: Do not just repeat the title. Please rearrange.
Response:
Done.
- Introduction: (1) There were too many paragraphs in this section, and the logical structure needed to be reorganized. (2) Please add to previous research on the environmental processes of CIP and TRI in soils. (3) The innovations of this paper should be clarified in the introduction section. (4) Line 42-47, it was not clear that how antibiotics reached into the soil. The sludge containing antibiotics was used directly for soil amendment? Livestock raising might be an even bigger source of antibiotic pollution?
Response:
(1) The introduction of the manuscript has been reorganized.
(2) The existing information has been completed with data on amounts of antibiotics found in soils, which indicates that these compounds are transferred from wastewater from treatment plants to soils (Lines 78-80).
(3) We explain the innovations of this paper in lines 88-91.
(4) Lines 42-47 (now lines 45-48) have been modified according to both questions from the reviewer.
- Materials and methods: (1) It was needed to explain how the sampling points of the 17 agricultural soils were selected. What were the representative of these soils? (2) It should be clarified that were there CIP and TRI detected in the soil sample. (3) Line 139-153, the author should also provide the relevant content of Quality Control, e.g., recoveries, method detection limits……
Response:
- Done (lines 99-100).
(2) The amounts of antibiotics in these soils are below the detection limit.
(3) Done. The quantification limits have been added (lines 159-160).
- Results: (1) Figures and Tables should be redrew, and it was hard to observe the differences between different soils or different antibiotics in current version. (2) The paper applied Freundlich, Langmuir and Linear models to describe the adsorption data of CIP and TRI. Which model had the better performance? Why?
Response:
- Initially we considered displaying the data for both antibiotics in a single table, but due to the large number of samples we saw that it was impractical.
(2) Done at lines 237-238, 254-255 and 285-287.
- Discussion: (1) Current version was hard to understand the mechanisms of adsorption/desorption. It seemed that only the electrostatic interaction was discussed in the Discussion section. How about other adsorption mechanisms? e.g., π-π electron donor-acceptor (EDA) interactions, H-bonding, and electrostatic effects. This section should be improved. (2) The results of multiple regression analysis did not perform well, and the R2values were very low. Other statistical models should be applied to further discuss these results. (3) Some soil physicochemical properties (pH, TSN, Al……) were shown to be correlated to antibiotic adsorption. It should be further discussed the mechanisms?
Response:
- Done at lines 346-352, 354-356, and 370-372.
- The regression equations presented are all statistically significant judging by their F values. We have found that, when establishing pedo-transfer functions, it is quite common to obtain R2 values similar to those presented by us in this work (Conde-Cid et al. 2020). On the other hand, we agree with the Reviewer that other models could be used, but nevertheless these types of models are very simple and allow an easier comparison with the data obtained by other authors, since they are more widely used.
Conde-Cid, M., Fernández-Calviño, D., Núñez-Delgado, A., Fernández-Sanjurjo, M.J., Arias-Estévez, M., Álvarez-Rodríguez, E., 2020. Estimation of adsorption/desorption Freundlich’s affinity coefficients for oxytetracycline and chlortetracycline from soil properties: Experimental data and pedotransfer functions. Ecotoxicology and Environmental Safety 196, 110584.
- Writing norms of academic papers should be rule and standard. For example, the parameters format was not unified, and the use of italic was improperly in this paper.
Response:
Corrected.
Reviewer 2 Report
The manuscript deals with a very important problem, which is the presence of emerging antibiotic-type pollutans in environmental systems. Therefore, the manuscript deals with a topic of global interest and highly publishable. Before being published, authors must make the following changes.
1. It is important that the introduction mentions the problem of emerging contaminants, unexpected effects of the two studied contaminants, which is reported in the literature with reference to the adsorption processes of these antibiotics. I suggest citing these articles found on the Web, they will help you enrich the introduction of your work: https://doi.org/10.1016/j.eti.2021.101589, , 10.1016/j.biortech.2020.122812, 10.1007/s11356-020- 10972-0, https://doi.org/10.1016/j.jwpe.2022.102582
2. Line 112, Table 1. Improve the structure of the two. On the other hand, in the structure of ciprofloxacin it is poorly drawn, since the secondary amine of the piperazinic ring should not be initially protonated. This means that it should not be found as a primary and charged amine.
3. In table 1, please change 290.32 to 290.32
4. In the methodology section, please describe: Location from where the soils were extracted (location), sampling method, procedure for treating them.
5. In the quantification of antibiotics, what are the limits of detection and quantification.
6. Please, in table 7 and 8, it is not clear for which types of soil the relationships of the desorption parameters with the edaphic variables are being evaluated, please make this clear
7. Based on point 6, please describe what is the function and the comparative effect of the realization of table 7 and 8.
8. From page 14 to page 20, I recommend placing some bibliographic references that explain or contribute to what is mentioned in these pages
Author Response
The manuscript deals with a very important problem, which is the presence of emerging antibiotic-type pollutants in environmental systems. Therefore, the manuscript deals with a topic of global interest and highly publishable. Before being published, authors must make the following changes.
- It is important that the introduction mentions the problem of emerging contaminants, unexpected effects of the two studied contaminants, which is reported in the literature with reference to the adsorption processes of these antibiotics. I suggest citing these articles found on the Web, they will help you enrich the introduction of your work: https://doi.org/10.1016/j.eti.2021.101589, 10.1016/j.biortech.2020.122812, 10.1007/s11356-020- 10972-0, https://doi.org/10.1016/j.jwpe.2022.102582
Response:
Done.
- Line 112, Table 1. Improve the structure of the two. On the other hand, in the structure of ciprofloxacin it is poorly drawn, since the secondary amine of the piperazinic ring should not be initially protonated. This means that it should not be found as a primary and charged amine.
Response:
Done.
- In table 1, please change 290.32 to 290.32
Response:
Done.
- In the methodology section, please describe: Location from where the soils were extracted (location), sampling method, procedure for treating them.
Response:
The location has been added and the sampling method and procedure is explained: lines 98-118.
- In the quantification of antibiotics, what are the limits of detection and quantification.
Response:
Done.
- Please, in table 7 and 8, it is not clear for which types of soil the relationships of the desorption parameters with the edaphic variables are being evaluated, please make this clear.
Response:
The relationships have been evaluated with all soils of the study, and the titles of tables 7 (adsorption parameters) and 8 (desorption parameters) have been modified for clarity.
- Based on point 6, please describe what is the function and the comparative effect of the realization of table 7 and 8.
Response:
Table 7 shows the correlations among edaphic variables and adsorption parameters, and Table 8 shows the correlations among edaphic variables and desorption parameters.
- From page 14 to page 20, I recommend placing some bibliographic references that explain or contribute to what is mentioned in these pages.
Response:
Done
Round 2
Reviewer 1 Report
This manuscript has improved a lot after revision. I agree to publish it in its current version.